# Electrocoalescence of Water Droplets

A. V. Shavlov *, V. A. Dzhumandzhi and E. S. Yakovenko

Institute of the Earth Cryosphere, RAS Siberian Branch, Tyumen 625000, Russia
* Correspondence: shavlov@ikz.ru

**Abstract:** An experimental setup has been created to study the electrocoalescence of submillimeter- and millimeter-sized water droplets on a hydrophobic dielectric surface. The dependences of the interdroplet distance on the droplet radius are studied. It is shown that drops on a hydrophobic surface exhibit patterns of spatial arrangement that are characteristic of drops of a droplet cluster and fog. The electric field strengths at which mass coalescence of droplets begin are measured. A new model of electrocoalescence based on the state diagram of a drop-ion plasma is proposed. The possible role of electrocoalescence in the problem of rapid rain formation in atmospheric clouds is discussed.

**Keywords:** droplet; droplet cluster; fog; cloud; droplet spatial ordering; electrocoalescence; rain formation

## 1. Introduction

Water droplets in atmospheric clouds and fog tend to be electrically charged. The origin of the droplet charge is associated with the absorption of light, medium, and heavy atmospheric ions by drops (ions are formed during radiation ionization of atmospheric gases and combine into complexes with neutral molecules), as well as with the transfer of protons and hydroxide ions between water and air during evaporation and condensation of drops. Light counterions contained in the interdroplet space, fully or partially compensating the charge of water droplets. In many cases, clouds and fog can be considered as drop-ion plasmas. Therefore, the processes occurring in clouds and fog are of particular interest from the point of view of plasma physics. This article is devoted to the study of the phenomenon of electrocoalescence of water droplets in the atmosphere, and the analysis of the mechanism of coalescence based on the phase diagram of the drop-ion plasma.

Micron-sized water droplets are formed in atmospheric clouds due to the condensation of water vapor on active condensation nuclei. Due to further condensation and diffusion of the vapor, the droplets can reach a radius of 10 μm. It is believed that drops with a radius of more than 50 μm acquire a sufficiently high fall velocity in the cloud, and grow by gravitational coalescence. The mechanism of droplet growth from 10 to 50 μm remains unclear. Observations show that the process of formation of raindrops in clouds requires only 15–20 min, while existing theories predict that the duration of droplet growth from a size of 10 to 50 μm should be a few to tens of hours [1–3]. In the scientific literature, the problem of the rapid growth of raindrops is known as "condensation-coalescence bottleneck in rain formation".

On the way to solving this problem, part of the research is focused on studying the increase in the velocity of droplets in a turbulent atmosphere and the turbulent-induced increase in the number of droplet collisions leading to their coalescence [1,4–7]. The results of a study of tangling clustering instability of small water droplets in a turbulent temperature stratified atmosphere [8] were promising, where the possibility of the formation of clusters with a drop concentration several orders of magnitude higher than the average concentration in a cloud was shown. Since the concentration and, accordingly, the rate of droplet coalescence within clusters increases sharply, the characteristic time of droplet

coalescence sharply decreases. This effect can increase in a stratified atmosphere [8,9]. Experimental verification of this mechanism is required.

Another part of the research on the way to solving this problem is focused on studying the possibility of coalescence of water droplets under the action of an electric field (electrocoalescence). Electrocoalescence has been studied for many years both theoretically and experimentally. Presumably, electrocoalescence is possible due to the dipole–dipole attraction of polarized water droplets in an external electric field. Many works report a significant effect of the electric field on the growth of raindrops [10–16]. Recent experimental studies [17] have shown a pronounced correlation between the size of raindrops falling on the earth's surface and the electric field strength in the cloud: the higher the field strength, the larger the droplet size. The mechanism of electrocoalescence requires additional experimental studies.

At present, apparently, it is impossible to give an objective preference to any of the indicated mechanisms of rapid rain formation. Further thorough theoretical and experimental studies are needed. Nevertheless, the authors of this work consider it promising to study the mechanism of electrocoalescence. This preference is based on the fact that electrocoalescence is easy to observe in simple domestic conditions. For example, one can observe how water droplets condense and grow on the inner walls of a hermetically sealed, transparent plastic bottle with a small amount of water. For two or three days of growth, the droplet radius can reach 1 mm or more, as in Figure 1a. Further, if you electrify, for example, a dielectric comb by rubbing against the hair and then bring the comb to the surface of the bottle, you can see how the drops coalesce with each other in a split second and form larger drops, as in Figure 1b. Such an experience is very impressive and can find many supporters of the electrocoalescence hypothesis.

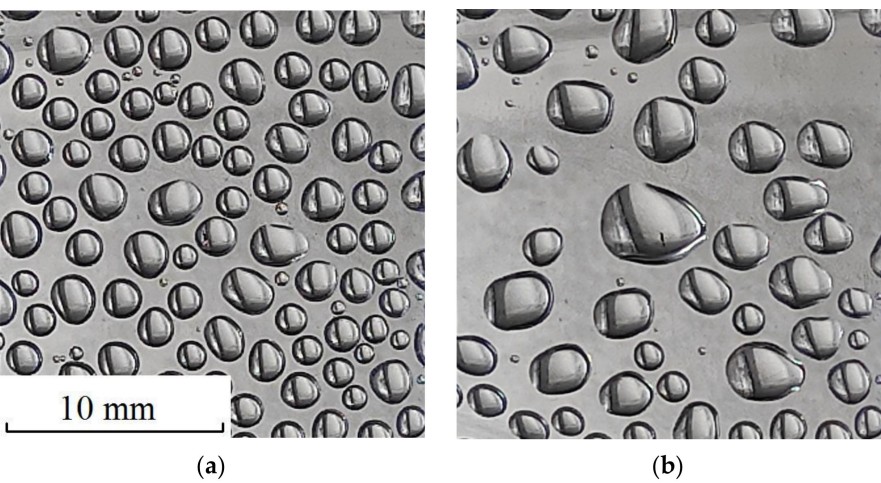

(**a**)　　　　　　　　　　　　　(**b**)

**Figure 1.** Drops of water on the wall of a plastic bottle before applying high voltage (**a**), after applying high voltage (**b**).

The purpose of this work was an experimental study of the patterns of electrocoalescence of water droplets on a hydrophobic dielectric surface. The hydrophobic surface had a minimal force effect on the droplets. At the same time, it fixed the position of drops in space, and made photo and video registration of drops convenient. Water droplets on a hydrophobic surface, as shown later in the article, showed the same signs of spatial order relative to each other as water drops in fog and possibly in a cloud. Therefore, water drops on a hydrophobic surface could be considered a rough, but nevertheless, physical model of a water mist or cloud, and one could hope to obtain useful objective experimental data on electrocoalescence that would help clarify the mechanism of the phenomenon.

## 2. Experiment Technique

Water drops were obtained on the inner surface of a transparent sealed plastic container. The inner surface of the container was successively washed with ethyl alcohol, bidistilled water and dried. The container was then half filled with bidistilled water, Figure 2. The volume of the container was about 5 L and the wall thickness was 0.5 mm. Small diurnal fluctuations in air temperature in the laboratory, amounting to several degrees relative to the standard temperature, caused water vapor to condense on the inner free surface of the container. Vapor condensation occurred on active condensation nuclei. There were apparently a lot of such nuclei on the surface, since after the drops were washed off with water, the spatial coordinates of the newly formed drops did not coincide with the coordinates of the drops that preceded them. The condensed moisture was collected in microscopic drops, since the plastic surface had pronounced hydrophobic properties compared to, for example, silicate glass. (The contact angle of the PET container material is about 70 degrees, while for silicate glass this angle is only 7 degrees.) Over time, the droplets increased in size due to condensation and occasional droplet pair coalescence. Drops could reach an average radius of about 2 mm in two or three days. The measurements were performed at various stages of droplet growth. The average droplet radius was recorded in the range of $r = [0.03, 2]$ mm.

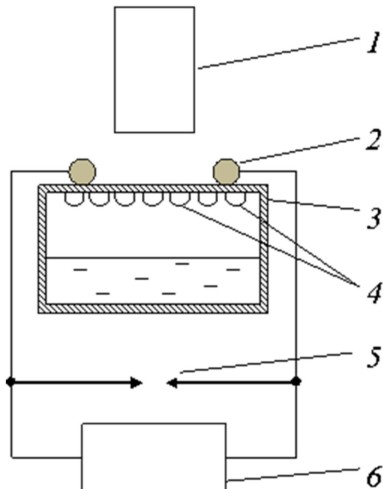

**Figure 2.** Block diagram of the installation. 1—camera, 2—cylindrical electrode, 3—plastic container with water, 4—drops of water, 5—discharger, and 6—electrostatic machine.

The position of drops on the surface was recorded with a camera of a Redmi Note 9 Pro device, Xiaomi with a resolution of $3472 \times 4640$ pixels, in the direction perpendicular to the surface. No additional lenses were used to enlarge the objects of observation. Droplets were photographed against the background of a measuring ruler with a division value of 1 mm. The droplet sizes and interdroplet distances in pixels were determined from the photographs using the Webbers ScopePhoto tool. Next, the pixels were converted to millimeters. The relative error in determining the geometric dimensions and distances was less than 10%. The error was due to the noticeable unsharpness (blurring) of the droplet boundaries in the photographs.

An electric field was applied to the system of drops outside the plastic surface using two parallel cylindrical electrodes pressed to the surface and covered with plastic insulation. The radius of each electrode together with the insulation was 1 mm, the length of the electrode was 20 mm, and the interelectrode distance was 20 mm. Voltage was applied to the electrodes from spherical (with a radius of 5 mm) balls of an air gap of an electrostatic (electrophore) machine. The potential difference created by the machine could reach 60 kV with a distance between the spark gap balls of at least 20 mm. The machine was put into operation for a time equal to 3–5 s. During this time, the voltage between the spark gap

balls reached the breakdown value several times, and electrical discharges occurred. The breakdown voltage amplitude $U$ was regulated using the air gap $\Delta$ between the spark gap balls and was estimated by the formula $U = \Delta \times E_0$, where $E_0 = 3 \times 10^6$ V $\times$ m$^{-1}$, which is the breakdown value of the electric field strength in air under atmospheric conditions close to standard (this is a fixed value). Thus, the voltage between the electrodes was sawtooth with a frequency of 1–10 Hz and adjustable in amplitude. According to the foregoing, the electric circuit between the electrodes consisted of (a) a near-electrode insulator layer (1 mm thick), (b) a system of drops on the surface (20 mm long), and (c) an insulator layer (1 mm thick) at the second electrode. The reactive resistance of the insulator layers at the moments of the electric discharge (with a duration of fractions of a millisecond) was estimated to be many orders of magnitude lower than the active resistance of the system of drops on the surface. Therefore, it could be assumed that at the instants of discharges, the entire voltage was applied to the droplet system. The amplitude of the field strength can be estimated in order of magnitude by the formula $E = U/d = E_0 \times \Delta/d$, where $d$ is the distance between the electrodes. In reality, the field between the cylindrical electrodes was inhomogeneous. Thus, at a distance of 3 mm from the electrode and at the middle of the distance between the electrodes, the field could differ by almost a factor of two. Therefore, the estimation of the electric field strength according to the indicated formula has a large systematic relative error of several tens of percent. This error is the same for all our experiments. The random relative error in determining the field strength was less than 10%. It depended on the errors in determining the quantities $E_0$, $d$, and $\Delta$. We believe that the random error in the value of $E_0$ is small (<10%), since the experiments were performed at atmospheric pressure and temperature close to the standard values; the random errors of the other two quantities also did not exceed 10%, since these quantities were determined by the same method as the droplet radii and interdroplet distances.

The technique for applying an electric field and photographic recording of a system of drops was as follows. First, droplets were photographed in a zero electric field (with closed spark gap balls). Then the air gap of the spark gap was increased (and, accordingly, the magnitude of the field applied to the drops) and the electrostatic machine was turned on for 3–5 s. After that, the system of drops and the gap of the spark gap were photographed. Then the spark gap was closed to remove the residual voltage. Further, the gap of the spark gap was increased even more, the electrostatic machine was turned on and off, photographic recording was performed, the spark gap was closed, and so on. An increase in the gap of the spark gap with subsequent photo registration was performed up to 10 times. Then, the drops remaining on the surface were washed off with the water contained in the container by shaking the container. Then, for several hours, or even tens of hours, spontaneous formation of the next system of droplets of the desired radius was expected. The entire cycle of electrical measurements with a tenfold change in the gap of the spark gap took about 5 min. During this time, no noticeable changes in droplet size due to condensation (evaporation) processes were observed. Also, during this time, no spontaneous coalescence of droplets was observed when the electric field was turned off.

### 3. Experimental Results

A system of droplets was photographed at various stages of their growth on a hydrophobic surface under atmospheric conditions close to standard. In each photograph, the shape of the droplets was close to round and the size of the droplets was approximately the same (the largest drop in the picture differed in size from the smallest by less than 2 times). With the passage of time (during two or three days from the beginning of growth), the average droplet radius increased. Figure 3 shows the dependence of the measured distance $\delta$ between the surfaces of neighboring drops on their average radius $r$, as well as the dependence of the calculated distance between the centers of neighboring drops $L = 2r + \delta$ on their average radius $r$. Averaging was performed over 5–10 pairs of drops. The trend of the distance between the surfaces of drops $\delta$ is described by a function $\delta = 0.025 \times r^{0.64}$ with a confidence value of $R^2 = 0.93$, where $\delta$ and $r$ are measured in meters. Below, this function

will be useful for analyzing the mechanism of droplet ordering on the surface. The trend of the distance between the centers of drops $L$ is described by a function $L = 1.35 \times r^{0.92}$ with a confidence value of $R^2 = 0.998$. For droplet radii in the range $r = [0.03, 2]$ mm, the distances between the centers of neighboring drops is, on average, 20% larger than their average diameter.

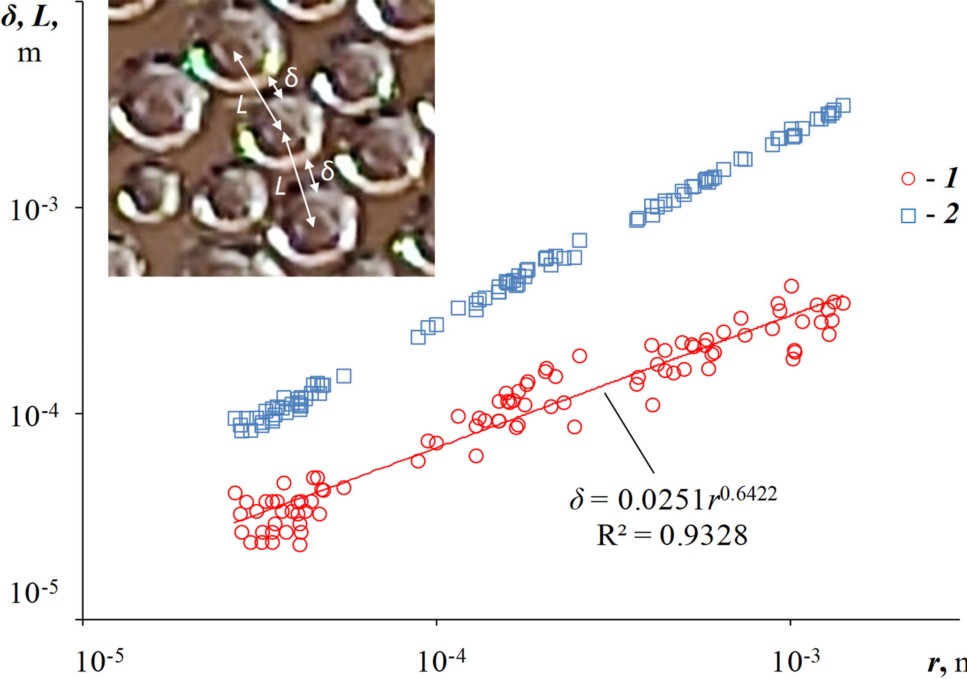

**Figure 3.** Dependence of the average distance between the surfaces of neighboring drops $\delta$ (1) and the average distance between the centers of neighboring drops $L$ (2) on their average radius $r$. The inset shows a fragment of a photograph of the droplet structure.

The electric field affected the droplet coalescence in the following way. So, for example, for droplets with a radius of 0.1 mm, coalescence was not observed in an electric field with a strength less than $0.2 \times 10^6$ V $\times$ m$^{-1}$. At a slightly higher field strength, separate pairs of droplets coalesced, located one after the other along the field strength lines. That is, coalescence occurred due to dipole–dipole attraction. With a further increase in tension, ensembles of three, five, or more drops coalesced. The drops in the ensembles were closely spaced around a single geometric center. In all cases, coalescence occurred in a fraction of a second. At even higher field strengths, the probability of coalescence decreased. The reason for this was that the drops became few, and they were at too great a distance from each other.

In order to numerically characterize the degree of influence of the electric field on coalescence, the concentration of droplets on the surface was determined at the lowest field strength, then at a higher, still higher, and so on. The concentration was determined as the ratio of the number of droplets falling into the field of view of the camera to the area of the field of view of the camera. In relative units, the initial concentration (that is, the concentration at the lowest field strength) was assumed to be 1. With each increase in field strength, the droplet concentration decreased due to coalescence. Figure 4 shows the dependences of the droplet concentration in relative units on the electric field strength for drops with initial average radii of 0.06 mm (curve 1) and 0.16 mm (curve 2). Curves 3 and 4 approximate curves 1 and 2, respectively. It can be seen from the graphs that droplets with an initial large size coalesce at lower electric field strengths than droplets with an initial small size. The critical value of the field strength, above which mass coalescence was observed, was determined from the projection onto the $E$-axis of the intersection point of two tangent lines drawn, for example, to curve 4 in Figure 4. Tangent lines were drawn

to the initial horizontal section of the curve and to the section with the maximum slope $|dN/dE|$. Figure 5 shows the dependence of the critical field strength on the average initial droplet radius. The critical field strength decreases with an increase in the average radius according to a law close to $E_c \sim r^{-1/2}$. The obtained dependence will be required further when discussing the mechanism of coalescence.

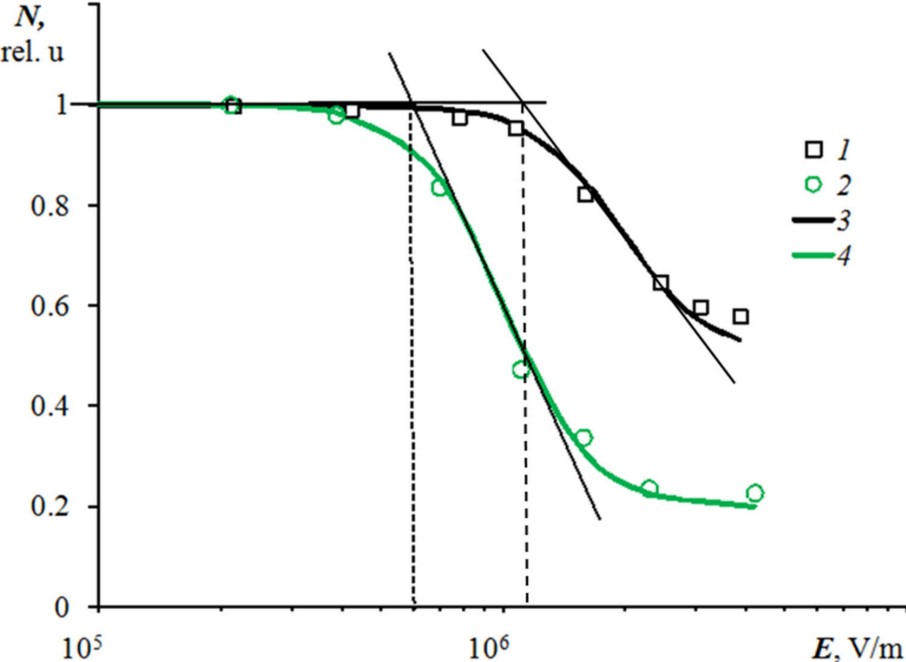

**Figure 4.** Dependence of the concentration of drops on the surface *N* on the strength of the external electric field *E*. (1)—the average radius of the drops is 0.06 mm, (2)—0.16 mm. (3) and (4) are approximating curves for (1) and (2), respectively.

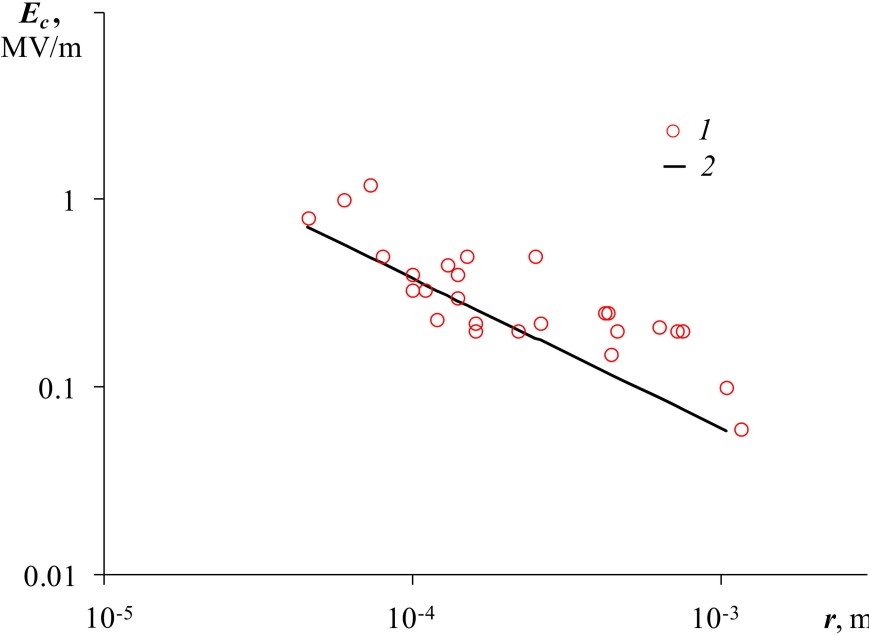

**Figure 5.** Dependence of the critical electric field strength $E_c$ on the average droplet radius *r*. (1)—experimental values and (2)—theoretical curve.

In high electric fields, the droplets, which had reached a sufficiently large size due to coalescence, could slowly slide over the plastic surface. Sliding was carried out in the direction of the nearest electrode to a region with a higher field strength (this behavior is typical for dielectrics and conductors in a nonuniform electric field). The exception was single drops that could move in the opposite direction. Apparently, these drops had a large free electric charge of their own and moved in accordance with the sign of the charge and the direction of the field.

## 4. The Discussion of the Results

### 4.1. Ordering of Drops on a Hydrophobic Surface

In the Introduction of this article, we promised to show that water drops on a hydrophobic surface show the same signs of spatial ordering relative to each other as water drops in a droplet cluster, in a fog, and possibly in a cloud. First, let us recall what signs of an ordered position of drops a droplet cluster and fog have. A droplet cluster spontaneously forms above the surface of heated water and is a flat structure in which drops are hexagonally ordered relative to each other [18]. Unlike a cluster, a fog consists of structural elements randomly located in space [19]. Structural elements are drop chains, polygons and more complex spatial formations in which drops are at the same distance from each other. The distance between the surfaces of the drops in the structural elements of the fog, as well as in the droplet cluster, obeyed the regularity [20].

$$\delta \approx 10^{-5} \times Z^{1/3}, \tag{1}$$

where $\delta$ is the distance between the surfaces of the drops in meters, $Z$ is the charge of the drop in units of elementary charge. Let us clarify that in a drop cluster, the value of $\delta$ was determined as the average value of the distance between the surfaces of neighboring drops over all drops of the cluster [20,21]. At the same time, the cluster could contain up to 100 drops. In fog, the value of $\delta$ was determined as the average value of distances within one structural element containing about 10 drops [19]. The charge of a drop in a drop cluster was determined from the frequency of the sound oscillations of the drop [21]. Next, the charge value was averaged over several drops. The charge of the drop in the structural element of the fog was estimated from the empirically established in [21] relationship between the charge and the radius of the drop. The corresponding formula will be given below. Thus, regularity (1) is substantiated by a sufficiently large number of measurements.

Dependence (1) agrees well with the state diagram of a drop-ion plasma published in [20]. In this paper, we considered a drop-ion plasma consisting of like-charged drops with charge Z, singly charged counterions compensating the charge of the drops, and a small number of singly charged ions. The state diagram is shown in Figure 6 with some modifications. Curve 1 depicts the interdroplet distance $\delta_{F\max}$ at the points of the local maximum of the free energy of all plasma particles depending on the droplet charge Z. The position of the curve does not depend on the radius of the droplets (free energy takes into account the gas-kinetic and electrostatic energies of the particles.) Curve 1 is well approximated by curve 5, which can be obtained by numerical calculation from the condition that the electrostatic and kinetic energies of all plasma particles are equal. This dependence is described by the formula $\delta_{F\max} \approx 10^{-5} \times Z^{1/3}$.

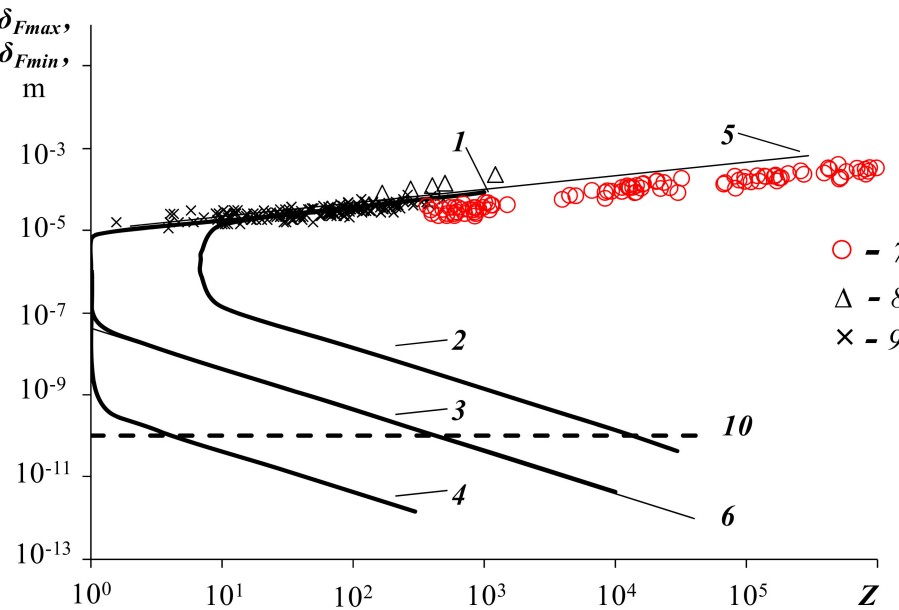

**Figure 6.** Diagram of states of a drop-ion plasma. (1) is the distance between the surfaces of the nearest drops $\delta_{F\max}$ at the points of the local maximum of free energy depending on the charge of the drops $Z$. (2), (3), and (4) are the distance between the surfaces of the drops $\delta_{F\min}$ at the points of the local minimum of the free energy, the radii of the drops are $10^{-5}$, $10^{-6}$ and $10^{-8}$ m, respectively. (5) and (6) are approximating curves. (7)—experimental values for drops on a hydrophobic surface, (8)—the same for a drop cluster, and (9)—the same for water fog. (10) is the distance between the droplet surfaces at which coalescence occurs.

Curves 2, 3, 4 in Figure 6 show the inter-droplet distance $\delta_{F\min}$ at the points of local minimum of the plasma free energy depending on the droplet charge $Z$ at droplet radii of $10^{-5}$, $10^{-6}$, and $10^{-8}$ m, respectively. These curves are approximated by the empirical Dependence (6) described by the formula

$$\delta_{F\min} \approx r^{1.2} \times Z^{-1}. \tag{2}$$

This formula will be required below to analyze the mechanism of droplet electrocoalescence. Note that the region of states located between curve 1, on the one hand, and curves 2, 3, 4, on the other hand, is the region of plasma attraction as a whole.

Markers 8, 9 in Figure 6 represent the experimental points determined in [21,22] for drops of a droplet cluster and water fog, respectively. It follows from Figure 6 that the experimental points for the cluster and fog are located at the upper boundary of the plasma attraction region. Probably, due to attraction, these objects exhibit spatial drop ordering. Why did the experimental points fall on the upper boundary of the plasma attraction region, and not on the more energetically favorable lower boundary corresponding to the local minimum of free energy? This may be due to the difficulties in dissipating the thermal energy of the plasma. Indeed, when passing from the upper boundary of the attraction region to the lower boundary, the plasma density should increase. In this case, the temperature will increase and, as was shown in [20], the plasma leaves the attraction region and, conversely, begins to expand. Therefore, the plasma state balances near the upper boundary of the attraction region. Problems with the dissipation of thermal energy probably do not arise when individual (rare) pairs of drops approach and pass to the lower boundary. This issue requires further study.

Let us show further that the position of drops on a hydrophobic surface also obeys the law of ordering of fog drops and drops of a droplet cluster. Let us use the empirically

established in [21] relationship between the charge of a drop and its radius in a fog and a droplet cluster in the form

$$Z \approx 10^{12} \times r^2. \tag{3}$$

(The origin of the droplet charge is associated with the exchange of water molecules, protons, and hydroxide ions between water and steam during evaporation/condensation [23]. The application of Formula (3) for drops on a hydrophobic surface is justified by the close physical conditions for obtaining drops to those conditions that were used to obtain a cluster and fog. In particular, the evaporation/condensation conditions were comparable at air humidity close to 100%.) Let us substitute Expression (3) into the experimentally established dependence for droplets on a hydrophobic surface $\delta = 0.025 \times r^{0.64}$ and obtain the dependence

$$\delta = 3.6 \times 10^{-6} \times Z^{0.32}. \tag{4}$$

Dependence (4) is shown in Figure 6, round markers 7. We see that Dependence (7) is in good agreement with Dependences (8) and (9) for the cluster and fog and is located close to the upper boundary of the plasma attraction region as a whole. Thus, the law of the mutual arrangement of drops on a hydrophobic surface coincides with the law of the arrangement of drops in fog.

### 4.2. Mechanism of Electrocoalescence

We assume that the coalescence of water drops is possible when the distance between the surfaces of the drops becomes less than the interatomic distance equal to $\delta_0 \approx 10^{-10} m$. At the same time, interatomic forces come into play and ensure coalescence. Let us consider the mechanism of coalescence based on the state diagram of drop-ion plasma, Figure 6. According to the diagram, single droplets of submicron size with a charge $Z > 3$ upon transition from the upper boundary of the plasma attraction region (curve 1) to the energetically more favorable lower boundary (curve 4) find themselves in the zone of distances between the droplet surfaces less than $\delta_0$. These droplets coalesce. (The level of interdroplet distances $\delta_0$ is shown in the plot of curve 10.) Drops of micron size (curve 3) and drops with a radius of $10^{-5}$ m (curve 2) can coalesce, passing to the lower limit of attraction, at charges over 300 and $10^4$, respectively. At lower charge values, coalescence is impossible. But, electrocoalescence is possible. Indeed, in an external electric field, water droplets are polarized and can acquire a large surface electric charge sufficient for interdroplet distances to fall into the region $\delta < \delta_0$ and coalescence to occur.

Let us perform a numerical estimate of the electric field strength required for the electrocoalescence of droplets of different radii and compare the results of the estimate with experimental data. Let us assume that the droplets are electrically conductive. The electric field strength inside the drop is zero. There is no dielectric polarization inside the drop. An electric field with strength $E$ induces a surface charge on the surface of a conducting ball (droplet) with a density [24]

$$\sigma = \frac{3\varepsilon_0 E}{4\pi} \cos(\theta), \tag{5}$$

where $\theta$ is the zenith angle of spherical coordinates, $\varepsilon_0$ is the electrical constant. (In the calculations, we choose a hemispherical drop model, which is a rather rough approximation. As a result, one can hope to obtain theoretical estimates of the drop parameters only by an order of magnitude). We integrate (5) over the surface of a hemisphere of radius $r$ and obtain the value of the charge induced by the field

$$Z = \frac{3\varepsilon_0 E r^2}{4e}, \tag{6}$$

where $e$ is the elementary charge. We substitute Expression (6) into (2) and obtain an expression for the critical field strength $E_c$ at which coalescence begins.

$$E_c = \frac{4e}{3\varepsilon_0\delta_0 r^{0.8}}.$$

(7)

According to this expression, the critical field strength at which coalescence begins is the smaller, the larger the droplet radius. On Figure 5 (curve 2) graphically shows the theoretical Dependence (7). It can be seen from Figure 5 that this dependence is close in slope and absolute values of the field strength to the experimental dependence (curve 1).

Thus, the mechanism of electrocoalescence of water droplets on a hydrophobic surface proposed in this article can withstand a simple numerical test and seems to be quite convincing. The study of this mechanism can serve to further advance the understanding of the nature of the growth of water droplets in fog, and the reasons for the rapid formation of raindrops in atmospheric clouds.

### 4.3. Experimental Estimation of the Droplet Charge on the Surface

Above, we used Formula (3) to estimate the droplet charge on the surface. Let us now estimate the droplet charge on the surface on the basis of experimental observations and compare it with the estimated result obtained using Formula (3). Thus, we will verify the correctness of such an estimate. When describing the behavior of droplets in an electric field, we mentioned that drops of large (millimeter) size could slide over a hydrophobic surface under the action of an electric field. Sliding was carried out in the direction of the nearest electrode to the region with a higher field strength. However, some drops could move in the opposite direction. We believe that the first drops behaved like a polarized body drawn into a region with a high field strength. The second drops behaved like free-charged particles in an electric field. Since we observed both drops in the same experiment, we can roughly assume that both forces are comparable in order of magnitude.

The force of drawing an electrically conductive ball into a region with a high field strength can be calculated by the formula

$$F = \int_S dF = \int_S E(\theta)\sigma(\theta)dS,$$

(8)

where $E(\theta) = E + \nabla E \times r \times \cos(\theta)$ is the inhomogeneous electric field around the ball, $E$ is the average value of the field strength, $\nabla E$ is the field gradient, $r$ is the radius of the ball, $\theta$ is the zenith angle of the spherical coordinates placed at the center of the ball, $\sigma(\theta)$ is the surface charge density of the ball, defined by the Formula (5), and $dS$ is the surface element of the ball. We integrate (8) over the surface of the ball and obtain

$$F = \varepsilon_0 r^3 E \nabla E.$$

(9)

The force acting on a free charge in an electric field is determined by the formula

$$F_Z = eZE.$$

(10)

Let us equate forces (9) and (10), and obtain a formula for estimating the electric charge of a drop based on experimental data

$$Z = \frac{\varepsilon_0 r^3}{e}\nabla E.$$

(11)

The experimental conditions under which the movement of the main part of the drops in the direction of the nearest electrode, and the small part in the opposite direction were observed, are as follows: $r = 2.5 \times 10^{-4}$ m, $\nabla E \approx \frac{E}{l} = 10^8$V $\times$ m$^{-2}$, where $E = 10^6$ V $\times$ m$^{-1}$—field strength in the experiment and $l = 10^{-2}$ m—half of the interelec-

trode distance. We use these data in Formula (11) and get $Z = 9 \times 10^4$. Let's compare this value with the value calculated by Formula (3): $Z = 10^{12} \times r^2 = 6 \times 10^4$. We see that the values coincide in order of magnitude. Thus, the use of Formula (3) for an approximate estimate of the charge of droplets on a hydrophobic surface is admissible.

## 5. Conclusions

Let us summarize the results achieved in this work.

1.  An experimental setup has been created to study the spatial ordering and electrocoalescence of water droplets on a hydrophobic plastic surface. An electrostatic (electrophore) machine with a voltage of up to 60 kV was used as a source of high voltage.
2.  The dependences of the average distance between the surfaces of the nearest drops on their average radius are studied. It has been established for the first time that drops on a hydrophobic surface exhibit patterns of spatial arrangement characteristic of a drop cluster and water fog. That is, the system of drops on the surface can serve as a good physical model of both. This circumstance allowed us to hope to obtain useful experimental data on electrocoalescence in fog using a system of droplets on the surface.
3.  The dependences of the number of coalesced droplets on the surface on the strength of the external electric field were measured. The dependences of the field strength at which mass coalescence begins on the average droplet radius in the range of average radii of [0.03, 2] mm are studied. A new model of droplet electrocoalescence based on the state diagram of a droplet-ion plasma, numerically consistent with experimental data, is proposed. Since drops on a hydrophobic surface are a physical model of a droplet cluster and fog, we hope that the pattern of electrocoalescence established by us can be suitable for understanding the phenomenon of coalescence in fog and clouds. Further numerical calculations of the growth rate of droplets in clouds, taking into account electrocoalescence, can put an end to the problem of "bottleneck condensation-coalescence in the formation of raindrops."

The results obtained in this work can contribute to a targeted search for "electrical" technologies for controlling, for example, the intensity of precipitation. This would be especially important for agricultural lands or forests that suffer from a lack of soil moisture during dry periods. Also of interest, would be technologies for forecasting heavy precipitation, for example, in cities, based on data from monitoring the electrical state of clouds and information from the drop-ion plasma state diagram. Other applications of the results obtained in this work are also possible. For example, they could be used in the modern production of microelectronics to control and reduce the intensity of the generation and growth of dust particles polluting the plasma during the plasma-chemical formation of microprocessor crystals. The obtained knowledge may also be of interest for the synthesis of nanosized powders from aerosol droplets of metal salts, and so on.

**Author Contributions:** Conceptualization, A.V.S. and V.A.D.; methodology, A.V.S. and V.A.D.; investigation, A.V.S., V.A.D. and E.S.Y.; writing, A.V.S. All authors have read and agreed to the published version of the manuscript.

**Funding:** The work was carried out according to state assignment 121041600047-2.

**Institutional Review Board Statement:** Not applicable.

**Informed Consent Statement:** Not applicable.

**Data Availability Statement:** The data presented in this study may be available upon request.

**Conflicts of Interest:** The authors declare that they have no known competing financial interests or personal relationships that could have appeared to influence the work reported in this paper.

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
