# Peer review of "Electrocoalescence of Water Droplets"

_plasma, doi:10.3390/plasma6010011_

Round 1
Author Response
Electrocoalescence of Water Droplets
A.V. Shavlov, V.A. Dzhumandzhi, E.S. Yakovenko 2
Institute of the Earth Cryosphere, RAS Siberian Branch, P.O. 1230, Tyumen 625000, Russia 3
E-mail: shavlov@ikz.ru
In the manuscript, the authors report their investigation on droplet coalescence under electric fields. Also, some numerical calculations are carried out to predict the phenomena they acquired. The model they proposed provides new ways for the following researchers to study a similar problem. I have some scientific concerns as shown below, and I am happy to recommend the paper be published after the concerns are fixed.
[1]. Line 86: the radius of r = 1÷2 mm ? Is it an average value? The drop size should be a distribution. The authors are suggested to type 0.5mm, instead of using ÷, for more readers to follow easily.
[2]. Line 90: Did the authors use a lens to magnify the observation objects?
[3]. Line 97: Once the images are captured, the process should not introduce quite larger errors. Could the author accurately point out how large the error is and where these errors are from?
[4]. Line 98: the machine was operated 0.6s? or 3~5 s?
[5]. Line132: it is not easy to understand 30÷10%, the authors are suggested to change to percentage.
[6]. Fig 3: The distance between the surfaces of the droplet pair is much smaller than that of the droplet. That’s why there is a larger deviation of δ while the deviation for L is much smaller. I guess the position where the droplet generates depends on the surface properties of the surface, like the nuclear cores for droplets to generate. How do the authors prepare the container surface before experiments? Also, the relation shown in Figure 3 is time dependent I guess, as the distance is relevant to the evaporation, and the coalescence. Would the authors describe in detail how you measure the distance by adding some figures of the raw image to fig 3? And also it will be better if the authors could discuss the dependence of the relation on the coalescence and evaporation.
[7]. Line 150: How is the droplet concentration defined?
[8]. Fig 4: I guess the authors run experiments continuously, for example, the square for case 1, initially N=1 , and E is applied to make N = 0.8 and then the procedure is repeated. While the process does not only change the droplet concentration, the drop size and the distance between the droplet pairs also change. Maybe the difference is seen between the black and green only because the initial droplet size differs a lot. Could the authors discuss more on this part, in combination with figure 3?
[9]. Line182, could the author describe how the distance between the droplet surface and the drop charge are determined? Is the droplet distance δ an average value, or it is an averaged by many measurements. Also the same for Z.
[10]. Line 194, what is the definition of δFmax? Please write it in the main text.
[11]. Line 245, and 246. How do the authors keep the droplet size at a certain value (10-6 or 10-5 m) in their experiments?
Responses to the Reviewer's remarks
[1] A clarification was made that "droplets could reach an average radius of about 2 mm."
[2] A clarification was made that “additional lenses were not used to enlarge the objects of observation. Droplets were photographed against the background of a measuring ruler with a division value of 1 mm.
[3] A clarification was made that “The relative error in determining the geometric dimensions and distances was less than 10%. The error was due to a noticeable blurring of the boundaries of the drops in the photographs.”
[4] Corrected: 3⁓5 s.
[5] A change has been made: "the distance between the centers of neighboring drops is, on average, 20% greater than their average diameter."
[6] The inner surface of the container was successively washed with ethyl alcohol, bidistilled water and dried. Then the container was half filled with bidistilled water...”. “Vapour condensation took place on active condensation nuclei. There were apparently a lot of such nuclei on the surface, since after the drops were washed off with water, the spatial coordinates of the newly formed drops did not coincide with the coordinates of the drops that preceded them.
“The photo registration of the system of drops was performed at various stages of their growth ... Over time (over two or three days), the average radius of the drops increased.” The points in Fig. 3 were obtained at different time points from the start of growth. For clarification in Fig. 3, a fragment of a photograph of drops is added and the measured values are indicated.
[7] Added clarification: “Concentration was defined as the ratio of the number of droplets falling into the field of view of the camera to the area of the field of view of the camera. In relative units, the initial concentration (that is, the concentration at the lowest field strength) was assumed to be 1".
[8] Added clarifications:
"In relative units, the initial concentration (that is, the concentration at the lowest field strength) was assumed to be 1."
“The technique for applying an electric field and photographing a system of drops was as follows. First, droplets were photographed in a zero electric field (with closed spark gap balls). Then the air gap of the spark gap was increased (and, accordingly, the magnitude of the field applied to the drops) and the electrostatic machine was turned on for 3–5 seconds. After that, the system of drops and the gap of the spark gap were photographed. Then the spark gap was closed to remove the residual voltage. Further, the gap of the spark gap was increased even more, the electrostatic machine was turned on and off, photographic recording was performed, the spark gap was closed, and so on. An increase in the gap of the spark gap with subsequent photo registration was performed up to 10 times. Then, the drops remaining on the surface were washed off with the water contained in the container by shaking the container. Then, for several hours or even tens of hours, spontaneous formation of the next system of droplets of the desired radius was expected. The entire cycle of electrical measurements with a tenfold change in the gap of the spark gap took about 5 minutes. During this time, no noticeable changes in droplet size due to condensation (evaporation) processes were observed. Also, during this time, no spontaneous coalescence of droplets was observed when the electric field was switched off.»
“From the graphs (Fig. 4) it can be seen that droplets with an initial large size coalesce at lower electric field strengths than droplets with an initial small size.”
[9] An explanation was added that “in a drop cluster, the value of δ was determined as the average value of the distance between the surfaces of neighboring drops over all drops of the cluster [20, 21]. At the same time, the cluster could contain up to 100 drops. In fog, the value of δ was determined as the average value of distances within one structural element containing about 10 drops [19]. The charge of a drop in a drop cluster was determined from the frequency of the sound oscillations of the drop [21]. Next, the charge value was averaged over several drops (up to 10 drops). The charge of the drop in the structural element of the fog was estimated from the empirically established in [21] relationship between the charge and the radius of the drop. The corresponding formula will be given below. Thus, regularity (1) is substantiated by a sufficiently large number of measurements.”
[10] Added an explanation of the values of δFmax and δFmin in the main text.
[11] Answer: We waited until the droplets reached approximately the required size, and performed a photograph or a five-minute cycle of electrical measurements.
We thank the Reviewer for valuable comments.

Reviewer 2 Report
Dear authors,
Is the finding that droplets on the surface under study show signs of spatial order characteristic of droplet clusters and water mists a new discovery? Or is it rather the knowledge of the dependence of the mean distance between the surfaces of the nearest droplets on their mean radius?
Can the mechanism of droplet growth from a size of 10 to 50 μm, which was unclear, now be explained by the developed model? The issue is the problem of the "condensation-coalescence bottleneck in the formation of raindrops", which is well known in the scientific literature. Observations suggest that the process of raindrop formation in clouds takes only 15-20 minutes, while existing theories predict that the growth time of droplets with a size of 10 to 50 μm should be several to ten hours?
Could the measured dependence of the number of droplet coalescence on external electric field strength and the given dependence of the field strength at which droplet mass coalescence begins on their mean radius contribute to the intended induction of precipitation? This would be particularly important over agricultural land or forests that suffer from a lack of soil moisture during dry periods. Will it be possible to use the developed droplet electrocoalescence model based on the droplet ion plasma state diagram to predict heavy precipitation, e.g. in cities?
Do the results obtained in this work currently only contribute to a better understanding of the mechanism of coalescence of water droplets in the atmosphere (the droplet-ion plasma agrees numerically with experimental data)? Can electrocoalescence not only play an important role in the rapid formation of raindrops in atmospheric clouds, but can this phenomenon also be used practically in the future?
Author Response
Dear authors,
Is the finding that droplets on the surface under study show signs of spatial order characteristic of droplet clusters and water mists a new discovery? Or is it rather the knowledge of the dependence of the mean distance between the surfaces of the nearest droplets on their mean radius?
Can the mechanism of droplet growth from a size of 10 to 50 μm, which was unclear, now be explained by the developed model? The issue is the problem of the "condensation-coalescence bottleneck in the formation of raindrops", which is well known in the scientific literature. Observations suggest that the process of raindrop formation in clouds takes only 15-20 minutes, while existing theories predict that the growth time of droplets with a size of 10 to 50 μm should be several to ten hours?
Could the measured dependence of the number of droplet coalescence on external electric field strength and the given dependence of the field strength at which droplet mass coalescence begins on their mean radius contribute to the intended induction of precipitation? This would be particularly important over agricultural land or forests that suffer from a lack of soil moisture during dry periods. Will it be possible to use the developed droplet electrocoalescence model based on the droplet ion plasma state diagram to predict heavy precipitation, e.g. in cities?
Do the results obtained in this work currently only contribute to a better understanding of the mechanism of coalescence of water droplets in the atmosphere (the droplet-ion plasma agrees numerically with experimental data)? Can electrocoalescence not only play an important role in the rapid formation of raindrops in atmospheric clouds, but can this phenomenon also be used practically in the future?
Responses to the Reviewer's remarks
[1] It was clarified that “For the first time, it has been established that drops on a hydrophobic surface exhibit patterns of spatial arrangement characteristic of a drop cluster and water fog. That is, the system of drops on the surface can serve as a good physical model of both. This circumstance allowed us to hope to obtain useful experimental data on electrocoalescence in fog using a system of droplets on the surface.”
[2] It was clarified that “the regularity established by us for the electrocoalescence of droplets on the surface can be suitable for understanding the phenomenon of coalescence in fog and clouds. Further numerical calculations of the growth rate of droplets in clouds, taking into account electrocoalescence, can put an end to the problem of "bottleneck condensation-coalescence in the formation of raindrops."
[3] It was noted that “The results obtained in this work can contribute to a targeted search for “electrical” control technologies, for example, the intensity of precipitation. This would be especially important for agricultural lands or forests that suffer from a lack of soil moisture during dry periods. Also of interest would be technologies for predicting heavy precipitation, for example, in cities, based on data from monitoring the electrical state of clouds and information from the drop-ion plasma state diagram.”
[4] It is noted that “Other applications of the results obtained in the work are also possible. For example, they could be used in the modern production of microelectronics to control and reduce the intensity of the generation and growth of dust particles polluting the plasma during the plasma-chemical formation of microprocessor crystals. The obtained knowledge may also be of interest for the synthesis of nanosized powders from aerosol droplets of metal salts. And other applications are also possible.
We thank the Reviewer for valuable comments.
Authors.

Reviewer 3 Report
The paper concerns the issue of the Electrocoalescence of Water Droplets. Use the Journal template. The paper is not in the required template. The obtained results should be discussed in detail. It is recommended to further highlight the significance of their work and clarity their novelty and originality. A lot of self-citations about 20%.
Author Response
The paper concerns the issue of the Electrocoalescence of Water Droplets. Use the Journal template. The paper is not in the required template. The obtained results should be discussed in detail. It is recommended to further highlight the significance of their work and clarity their novelty and originality. A lot of self-citations about 20%.
Responses to the Reviewer's remarks
The article was designed in accordance with the template. In addition, the results obtained were discussed. Emphasized the significance and novelty of the work. We used only the most necessary self-citations.
We thank the Reviewer for valuable comments.

Reviewer 4 Report
Electrocoalescence of Water Droplets is very interesting paper not only concerning the problem of rapid rain formation but also in synthesis of nanosized powders from aerosol droplet of metallic salts. Some improvement are required.
Line 7: An experimental setup has been created to study the electrocoalescence of water droplets on a hydrophobic dielectric surface. In which range are water droplets (micron?submicron? Nanosized diameter?)
Line 16: The origin of the charge of drops is associated with the absorption of atmospheric ions by drops (which type of ions? Metallic ions)
Line 256: where θ is the zenith angle of spherical coordinates (the form of droplet is not spherical!) What is an error in results regarding this choise of spherical form!
Conclusion
Line 304: An electrostatic (electrophore) machine was used as a source of high voltage (please to write this value in kV)
Line 310: The dependences of the field strength at which mass coalescence of drops begins on their average radius are studied. The average radius vale shall be added in conclusion.
General questions:
1. What is temperature influence on electrocoascelence of water droplet
2. Did you measure surface tension and density of water droplet
Author Response
Electrocoalescence of Water Droplets is very interesting paper not only concerning the problem of rapid rain formation but also in synthesis of nanosized powders from aerosol droplet of metallic salts. Some improvement are required.
Line 7: An experimental setup has been created to study the electrocoalescence of water droplets on a hydrophobic dielectric surface. In which range are water droplets (micron?submicron? Nanosized diameter?)
Line 16: The origin of the charge of drops is associated with the absorption of atmospheric ions by drops (which type of ions? Metallic ions)
Line 256: where θ is the zenith angle of spherical coordinates (the form of droplet is not spherical!) What is an error in results regarding this choise of spherical form!
Conclusion
Line 304: An electrostatic (electrophore) machine was used as a source of high voltage (please to write this value in kV)
Line 310: The dependences of the field strength at which mass coalescence of drops begins on their average radius are studied. The average radius vale shall be added in conclusion.
General questions:
- What is temperature influence on electrocoascelence of water droplet
- Did you measure surface tension and density of water droplet
Responses to the Reviewer's remarks
[1] The annotation additionally indicated that the water droplets had millimeter and submillimeter sizes.
[2] It was clarified that “The origin of the charge of drops is associated with the absorption by drops of light, medium and heavy atmospheric ions (ions are formed during radiation ionization of atmospheric gases and combine into complexes with neutral molecules), as well as with the transfer of protons and hydroxide ions between water and air during evaporation and condensation of drops.
[3] In addition, we indicated that “In the calculations, we choose a hemispherical droplet model, which is a rather rough approximation. As a result of this, it will be possible to hope for obtaining theoretical estimates of the droplet parameters only by an order of magnitude.”
[4] The article additionally indicated that "The potential difference created by the machine could reach 60 kV with a distance between the spark gap balls of at least 20 mm."
[5] Additionally, the interval of values of the average radius [0.03–2] mm was indicated.
Answer to common questions:
The study of the effect of temperature on electrocoalescence, the measurement of surface tension, and the density of a water drop are questions that will be included in the plan of our further research.
We thank the Reviewer for valuable comments.
Authors.

Round 2
Reviewer 1 Report
Still some typos are found like 3 ÷ 5 s. The authors are suggested to proofread before publishing.
Author Response
Dear Reviewer, we have completed the proofreading.
Authors.
Reviewer 3 Report
Use the Journal template. The paper is not in the required template.
Plasma | Instructions for Authors (mdpi.com)
Author Response
Dear Reviewer, we have designed the article according to the template of the journal.
Authors.